# Isolation, NMR Characterization, and Bioactivity of a Flavonoid Triglycoside from *Anthyllis henoniana* Stems: Antioxidant and Antiproliferative Effects on MDA-MB-231 Breast Cancer Cells

**DOI:** 10.3390/antiox13070793

**Published:** 2024-06-28

**Authors:** Amani Ayachi, Guillaume Boy, Sonda Samet, Nathan Téné, Bouthaina Bouzayani, Michel Treilhou, Raoudha Mezghani-Jarraya, Arnaud Billet

**Affiliations:** 1Laboratory of Organic Chemistry LR17ES08, Natural Substances Team, Faculty of Sciences of Sfax, University of Sfax, P.O. Box 1171, Sfax 3000, Tunisia; amaniayachi21@gmail.com (A.A.); samet.sonda95@gmail.com (S.S.); bouzayanibouthaina@yahoo.com (B.B.); 2Equipe BTSB-EA 7417, Institut National Universitaire Jean-François Champollion, Université de Toulouse, Place de Verdun, 81012 Albi, France; guillaume.boy@univ-jfc.fr (G.B.); nathan.tene@univ-jfc.fr (N.T.); michel.treilhou@univ-jfc.fr (M.T.)

**Keywords:** *Anthyllis henoniana*, LC–MS/MS, phenolic compounds, flavonoids NMR, isorhamnetin-3-*O*-rutinoside-7-*O*-rhamnopyranoside, ROS, cell proliferation, cell viability

## Abstract

Plant extracts are considered as a large source of active biomolecules, especially in phytosanitary and pharmacological fields. *Anthyllis henoniana* is a woody Saharan plant located in the big desert of North Africa. Our previous research paper proved the richness of the methanol extract obtained from the stems in flavonoids and phenolic compounds as well as its remarkable antioxidant activity. In this research, we started by investigating the phytochemical composition of the methanol extract using high performance liquid chromatography coupled with electrospray ionization mass spectrometry (LC–MS/MS). Among the 41 compounds identified, we isolated and characterized (structurally and functionally) the most abundant product, a flavonoid triglycoside (AA_770_) not previously described in this species. This compound, which presents no cytotoxic activity, exhibits an interesting cellular antioxidant effect by reducing reactive oxygen species (ROS) generation, and an antiproliferative action on breast cancer cells. This study provides a preliminary investigation into the pharmacological potential of the natural compound AA_770_, isolated and identified from *Anthyllis henoniana* for the first time.

## 1. Introduction

Searches for novel biologically active compounds from plant sources are initiated by the careful selection of plants, followed by ethnobotanical studies. Studying plant extracts allows the development of more efficient processes for extraction and structural characterization of their bioactive compounds.

The genus *Anthyllis* includes 20 species growing in Europe, Africa, and the Mediterranean Basin. Previous studies proved the richness of *Anthyllis* extracts in phenolic compounds, especially flavonoids [1,2]. Our previous research article showed that the *Anthyllis henoniana’s* methanol extract from the stems harvested in May (Spring) presented the highest total phenolic and flavonoid contents (TPC and TFC) as well as remarkable antioxidant activity compared to that obtained from the stems harvested in February (Winter) [3]. 

Indeed, one of the main properties of phenolic compounds, especially flavonoids extracted from plant material, is the potent antioxidant activity responsible for neutralizing reactive oxygen species (ROS) [4,5]. However, in most phytobiological studies, effects of plant extracts on ROS production are assessed using chemical antioxidant tests, which are not representative of biological systems. Nevertheless, ROS are bioproducts of cellular metabolism, and their high accumulation in cells induces oxidative stress, leading to cell proliferation and apoptotic cell death [6]. Therefore, ROS are interesting targets in human therapeutics, particularly for limiting the proliferation of cancer cells [7]. Among all cancer types, breast cancer is the most prevalent worldwide. There are several subtypes of breast cancer, but triple-negative breast cancer is associated with poorer prognosis [8], and limited treatment options are available [9].

Therefore, there is considerable interest in identifying new compounds, and our previous results on *A. henoniana* extracts encouraged us to isolate biologically active compounds and test their potential biological activity on cancer cells. Thus, in this study, we provided a phytochemical screening of the *A. henoniana* methanol extract using LC–MS/MS analysis and proceeded with purification of the main compound (named AA_770_) using preparative HPLC. Finally, we tested several biological actions (i.e., cytotoxicity, ROS production, and cell proliferation and migration) of AA_770_ on breast cancer cells and compared the product’s potency with those of secondary metabolites of the same class. It is noteworthy to mention that this study is considered the first investigation to determine the chemical composition of *A. henoniana* methanol stem extract as well as structurally elucidate the biological activity of its purified compound.

## 2. Materials and Methods

### 2.1. Collection and Extraction of Plant Material

*Anthyllis henoniana* was collected in May 2021 from the Sahara of Beni Khedache Medenine Tunisia, approximate to this GPS coordination of 33°13′09.6″ N 10°12′46.2″ E. The botanical identification was carried out in the botany laboratory of the Faculty of Sciences, University of Sfax, Tunisia, by Dr. Zouhaier NOUMI, and a voucher specimen (LCO 140) was deposited at the herbarium of the Laboratory of Organic Chemistry (LR17-ES08), Natural Substances Team, Faculty of Sciences, University of Sfax. A mass of 500 g of dried stems was milled and placed into a cotton cartridge, and then extraction was performed successively with hexane, ethyl acetate, and methanol using a Soxhlet extractor (NS29/32 cone and NS 45/40 socket, Lenz, Germany). Two liters of each solvent were used. The temperature was raised to solvent boiling point and was maintained for 24 h.

Then, all three extracts were filtered and evaporated using a rotavapor at 40 °C to identify extraction yields. Thereafter, they were stored at 4 °C prior to analysis. 

### 2.2. LC–MS^n^ Analysis

The methanolic extract of *Anthyllis henoniana*’s stem was investigated using a Thermo Scientific LTQ XL mass spectrometer (Thermofisher, Courtaboeuf, France) fitted with an electrospray ionization source in the negative mode. Thermo roadmap software Xcalibur 4.4.16.14 was used to record negative ion spectra. A C_18_ reversed-phase Luna column at 30 °C (5 m, 150 mm 2.1 mm) was delivered to Surveyor HPLC for analysis. A: 0.1% formic acid in water (5% ACN), *v*/*v* and B: 0.1% formic acid in acetonitrile, *v*/*v*, were the selected solvents. The elution gradient was set from 0 to 40% of B for 40 min, 100% of B after 50 min, and the column was re-equilibrated between individual runs. The mobile phase had a flow rate of 0.2 mL·min^−1^, and the injection volume was 20 µL. The ion spray voltage was fixed at 3.5 V for the ESI source, and the capillary temperature was calibrated to 300 °C; the sheath and auxiliary gas pressures were set to 50 and 5 psi, respectively. The spectral range was from *m*/*z* 50 to 1200. The approach combined full scans and MS/MS experiments using a collision energy ranging from 10 to 35 eV, depending on the molecular masses of the compounds.

### 2.3. Isolation of the Major Compound Using HPLC-Preparative

The methanolic extract was first dissolved in 1 mL of methanol (100 mg·mL^−1^), and the solution was centrifuged at 3000 tr/min. The clear supernatant was collected, leaving behind a pellet of insoluble molecules. A volume of 100 µL of supernatant was injected on a HPLC–DAD chromatography system (Vanquish) to separate the major compound detected by mass chromatography.

Molecules were eluted at 3 mL·min^−1^ with an injection volume of 100 μL through a Betasil inverse phase C18 (10 × 250 mm, 5 μm, 100 Å) column using the following MS-grade solvents and reagents: A: aqueous buffer; B: acetonitrile. The solvent’s binary composition consisted of a linear concentration gradient from 0 to 23% of B for 23 min, 100% of B from 24 to 29 min, and 100% of A from 30 to 35 min. Fractions were collected in Eppendorf tubes every 30 s, in order to check their purity in LC–MS/MS thereafter.

### 2.4. Structural Identification of the Active Compound

The nuclear magnetic resonance spectra of the isolated product were recorded at the joint NMR service of Paul Sabatier University of Toulouse on a Brüker Avance 500 MHz device (frequencies of 500 (^1^H) and 125 MHz (^13^C)). The sample was solubilized in deuterated solvent CD_3_OD in a 5 mm diameter analytical tube. The chemical shifts (δ) are expressed in ppm, and the coupling constants are expressed in Hz. The software used to process the NMR spectra was MestReNovaTM 9.0 (Mestrelab Research, Santiago, Spain).

### 2.5. Spectroscopic Data

Compound AA_770_ (isorhamnetin-3-*O*-rutinoside-7-*O*-rhamnopyranoside), yellow amorphous powder, UV λ_max_ (MeOH) 256, 352 nm. ^1^H-NMR (500 MHz, CD_3_OD) δH ppm: 6.50 (1H, d, *J* = 2 Hz, H-6), 6.79 (1H, d, *J* = 92.5 Hz, H-8), 8.07 (1H, d, *J* = 2 Hz, H-2′), 6.98 (1H, d, *J* = 2.1, H-5′), 7.65 (2H, dd, *J* = 8.5, H-6′) 3.99 (1H, s, H-3). Glu: 5.32 (1H, d, H-1″), 3.84 (m, H-2″), 3.81 (d, *J* = 3.5 Hz, H-3″), 3.85 (m, H-4″); 3.68 (t, H-5″) 3.47 and 3.77 (m, dd, *J* = 10.4; 5.7 Hz). Rham-1: 4.55 (d, *J* = 1.5 Hz, H-1‴), 3.29 (t, H-2‴), 3.49 (m, H-3‴), 4.05 (dd; *J* = 5.5; 1.5 Hz, H-4‴), 3.55 (dd; *J* = 6.3, 2.9 Hz, H-5‴), 1.19 (3H, d; *J* = 6 Hz, H-6‴). Rham-2: 5.59 (d; *J* = 1.5 Hz, H-1⁗) 3.58 (m, H-2⁗), 3.58 (m, H-3⁗), 3.50 (m, H-4⁗), 3.61 (m, H-5⁗), 1.28 (3H, d; *J* = 6 Hz, H-6⁗).

^13^C-NMR (125 MHz, CD_3_OD): δC ppm: 158 (C-2), 134.3 (C-3), 178.3 (C-4), 161.4 (C-5), 99.2 (C-6), 162.4 (C-7), 94.2 (C-8), 156.6 (C-9), 105.8 (C-10), 121.3 (C-2′), 113.2 (C-2′), 149.5 (C-3′), 147.1 (C-4′), 114.7 (C 5′), 122.3 (C-6′), 149.8 (C-7′), 55.7 (OCH3). Glu: 103.1 (C-1″), 71.6 (C-2″), 68.6 (C-3″), 70.8 (C-4″), 74.2 (C-5″), 66.1 (C-6″). Rham-1: 100.5 (C-1‴), 72.4 (C-2‴), 70.8 (C-3‴), 70.3 (C-4‴), 68.3 (C-5‴), 16.6 (C-6‴). Rham-2: 98.5 (C-1⁗), 73.5 (C-2⁗), 70.8 (C-3⁗), 72.2 (C-4⁗), 69.9 (C-5⁗), 16.7 (C-6⁗).

### 2.6. Cell Culture

The triple-negative human breast cancer adenocarcinoma cell line MDA-MB-231 was generously gifted by Pr. Bystricky. Cells were maintained in high-glucose Dulbecco’s Modified Eagle’s medium (DMEM, Sigma-Aldrich, Saint Louis, MO, USA) supplemented with 10% fetal bovine serum (Grosseron, Coueron, France) and 1% penicillin/streptomycin (Sigma-Aldrich) and cultivated at 37 °C in a 5% CO_2_-humidified incubator. 

### 2.7. Cytotoxicity and Cell Proliferation Assays

The stock solutions of the compounds were prepared in DMSO with a maximum of 0.5% DMSO in final solution. The effect of compounds on cytotoxicity and cell viability were assessed using lactate dehydrogenase (Interchim, Montluçon, France) and Cell Counting Kit-8 (CCK-8; Sigma Aldrich, St. Louis, MO, USA) assays, respectively. Cells were seeded into a 96-well plate (5000 cells/well) and allowed to sit overnight before a 24 h exposure to 10 and 100 µmol·L^−1^ compound concentrations. Then, assays were performed following the manufacturer’s information, as previously described [10]. 

To evaluate the effects of AA_770_ and isorhamnetin on cell proliferation, MDA-MB-231 cells were plated on day 0 into a 96-well plate (5000 cells/well) and allowed to adhere for 24 h at 37 °C before exposure to the tested compound. Cell proliferation was measured from day 1 to day 5 using the Cell Counting Kit-8 (Sigma Aldrich) assay following the manufacturer’s instructions. Cell media were replaced and tested substances were renewed every day. 

### 2.8. Wound Healing Assay

MDA-MB-231 cells were plated into a 96-well plate (1 × 10^5^ cells/well) and allowed to grow until reaching 90% density. Then, a scratch was performed with a 10 µL pipette tip. Floating cells were removed by rinsing twice with PBS. Images of the wound were taken 2 h and 36 h after scratching. Then, the healing speed was assessed by measuring the wound area with Image-J software version 1.54 (NIH, USA).

### 2.9. Evaluation of Intracellular ROS Production by H_2_DCFDA Assay 

The 2′,7′-dichlorodihydrofluorescein diacetate (H_2_DCFDA) assay kit (Abcam, Cambridge, UK) was used to determine intracellular ROS levels in MDA-MB-231 cells. Briefly, cells were seeded into 96-well black plates (1 × 10^4^ cells/well) and incubated overnight. Cells were incubated for 45 min with 20 µmol·L^−1^ DCFDA at 37 °C, according to the manufacturer’s instructions. Then, the DCFDA was rinsed twice with provided buffer and immediately incubated with 30 µmol·L^−1^ TBHP (tert-butyl hyperoxide) to stimulate ROS production, with or without tested compounds at various concentrations. A condition without TBHP was carried out to measure basal ROS production in cells. After 4 h of incubation at 37 °C, the fluorescence emitted by the 2′,7′-dichlorofluorescein (DCF) product was measured using a fluorescence plate reader (Cytation 1, Biotek, Germany) at 485 nm for excitation and 535 nm for emission.

### 2.10. Statistics

Values are presented as mean ± S.E. of n experiments, as indicated for each experiment. Data were compared using the one-way ANOVA test following Tukey’s posttest. Differences were considered statistically significant at *p* < 0.05. All statistical tests were performed using GraphPad Prism version 8.0 (GraphPad Software, La Jolla, CA, USA).

## 3. Results and Discussion

### 3.1. Extraction

The extraction yields were calculated for each solvent (hexane, ethyl acetate, and methanol) and are shown in Table 1.

The obtained yields for the different extracts are as follows: 1.1% for hexane, 3.6% for ethyl acetate, and, lastly, 1.4% for methanol. These yield variations could be explained by the solvent polarity as well as the class of phytochemicals present [3].

### 3.2. LC–MS^n^

Our previous research article showed that the methanol extract from the stems harvested in May presented a high content of flavonoids (TFC) and phenolic compounds (TPC), as well a strong antioxidant activity [3]. Based on these results, LC–MS/MS analysis was conducted to determine the phytochemical composition of this extract. To identify the flavonoids aglycones, glycosides, steroids, anthocyanidin, chalcones, lignans, coumestans, pterocarpene, and phenolic acids present in this extract, we measured the loss of monosaccharides, disaccharides, and even trisaccharides, as well as their moieties. The results in Table 2 show a major compound detected at 21.17 min that we isolated using preparative HPLC.

### 3.3. Structure of the Major Compound Named AA_770_

#### 3.3.1. Purification on Preparative HPLC

Liquid chromatography coupled with UV–DAD and mass detectors enabled to separate the constituents of the extract (Figure 1a,b). For purification, we used a preparative column and only a UV–DAD detector, bearing in mind that the compound of interest has two absorption bands. The first band at 256 nm is characteristic of the A ring of the flavonoid, and the second band at 352 nm is characteristic of the B ring absorption of the flavonoid (Figure 1c). The purified compound was named AA_770_. In order to obtain 3 mg of pure AA_770_ compound required for the NMR studies, we carried out fifty purifications of 10 mg each, corresponding to a yield of 0.6%.

#### 3.3.2. Mass Analysis of AA_770_

The analysis of the AA_770_ with LC–MS/MS in negative mode (Figure 2) showed that we succeeded in isolating a pure compound and generated a pseudo-molecular ion at *m*/*z* 769. Its MS^2^ spectrum showed a daughter ion at *m*/*z* 623 due to the loss of a rhamnose. The MS^3^ mode generated many fragments: the first ion is at *m*/*z* 357 [623–266]^−^ corresponding to the loss of a deoxyhexose; the second major ion is at *m*/*z* 315 [357–42]^−^, resulting from the loss of the rest of the sugar C_2_H_2_O, is characteristic of the aglycon isorhamnetin; the latter evolved into the loss of a CH_3_ group to yield the ion at *m*/*z* 300; finally, a cleavage of the C ring caused the departure of a COH and the appearance of the fragment at *m*/*z* 271. The bibliography identifies this compound as isorhamnetin-glucosyl-di-rhamnoside and proposes its structure as an isorhamnetin aglycone comprising three sugars all linked in position 3 of the flavonoid (one hexose and two deoxyhexoses). Generally, during an LC–MS/MS analysis, we see a loss of three sugars when they are linked to each other. Therefore, it was interesting to confirm or refute this proposed structure via nuclear magnetic resonance.

#### 3.3.3. NMR Analysis of AA_770_

The examination of the ^1^H NMR spectrum at 500 MHz of compound AA_770_ dissolved in CD_3_OD showed that it is in favor of a flavonoid triglycoside, as we distinguished the presence of three anomeric protons corresponding to three sugar units (three doublets at δ 5.32 ppm (*J* = 7.8 Hz), 4.55 ppm (*J* = 1.5 Hz), and 5.60 (*J* = 1.5 Hz)). As for the aglycones part, we detected the presence of five doublets corresponding at δ = 6.50 ppm (*J* = 2 Hz), 6.79 ppm (*J* = 2.5 Hz), 8.07 ppm (*J* = 2 Hz), and 6.94 ppm (*J* = 2.1 Hz) corresponding to H6, H8, H2′, and H5′, respectively. The doublet–doublet at δ = 7.65 ppm (*J* = 8.5; 2 Hz) was attributed to H6′, and the singlet at δ 3,99 ppm corresponds to the protons of the methoxy group.

The *Jmod* ^13^C NMR spectrum carried out in CD_3_OD at 125 MHz showed the presence of 32 signals related to 32 types of carbons, with the presence of a characteristic signal at 178 ppm corresponding to the carbonyl group of the aglycone. The COSY H-H spectrum allowed us to determine the ^3^J_H-H_ homonuclear couplings. In order to confirm these attributions, we undertook a careful study of the 2D spectra (HSQC and HMBC). Therefore, the absence of correlation spots in HSQC allowed us to attribute the following quaternary carbons: C2 (158 ppm), C3 (134.25 ppm), C4 (178.30 ppm), C5 (161.39 ppm), C7 (162.38), C9 (156.64 ppm), C10 (105.83 ppm), C1′ (121.27 ppm), C3′ (149.68 ppm), and C4′ (147.06 ppm). The HMBC spectrum allowed us to observe a correlation spot between the anomeric proton of the hexose H1″ and carbon C3 of the aglycone and another correlation spot between the proton H1‴ of the deoxyhexose and the carbon C6″ of the hexose. On the spectrum, we also detected a correlation spot between the anomeric proton of the third sugar and carbon 7 of the aglycone, suggesting its bond at position 7. These results allowed us to pinpoint the exact positions of the three sugars.

To better confirm this structure, we used the NOSEY spectrum that allowed us to determine the couplings between protons correlating in space, as we observed correlation spots on either side of the symmetry axis (see NMR spectra in Appendix A). Regarding these results and the literature, we were able to find an exact attribution to the sugars, and the final structure of compound AA_770_ is shown in Figure 3.

### 3.4. AA_770_ Does Not Show Cytotoxicity on MDA-MB-231 Breast Cancer Cells

The cytotoxic activity of AA_770_ was evaluated on the breast cancer cell line MDA-MB-231 using the lactate dehydrogenase (LDH) leakage assay, which measures LDH activity in the extracellular medium as an indicator of irreversible cell death. As illustrated in Figure 4 (black bars), 24 h treatment with various concentrations of AA_770_ did not result in a significant difference in cytotoxicity compared to the negative control (DMSO). To confirm cell viability and normal cellular metabolism, a cell viability assay based on CCK-8/WST reduction was performed (Figure 4, grey bars). The results showed no significant difference in cell viability between the negative control (0.5% DMSO) and cells treated with various concentrations of AA_770_ after 24 h of incubation, indicating no harmful effect of the tested compound on cellular metabolism.

### 3.5. Cellular Anti-Radical Activity of AA_770_

To evaluate the biological interest of AA_770_, we first tested its effect on ROS production using the DCFDA fluorogenic probe. Unlike other chemical methods commonly used to measure antioxidant activity, this test was developed to be more biologically relevant. The anti-radical activity was demonstrated on MDA-MB-231 cells incubated with tert-butyl hyperoxide (TBHP), a well-known oxidative stress-inducing agent.

These results demonstrate a basal ROS level in MDA-MB-231 cells, which increased four-fold upon treatment with TBHP. Incubation with 10 µmol·L^−1^ of AA_770_ resulted in a slighter but statistically significant decrease in TBHP-induced ROS production, indicating its antioxidant activity.

Isorhamnetin is a flavonoid that has already shown an inhibitory ROS effect [38]. Taking into consideration its presence in the studied extract, it was interesting to also test its antiradical power and compare it with the one obtained for the isolated product AA_770_, which is an isorhamnetin derivative.

As shown in Figure 5, isorhamnetin leads to a more pronounced reduction in the TBHP-induced ROS levels compared to AA_770_, and effectively reduces ROS production to basal levels. In comparison, AA_770_ exhibited a weaker inhibitory effect on ROS, likely due to its structure as a triglycosylated derivative of isorhamnetin. The number and positions of hydroxyl groups, particularly at positions 3′ and 4′ on the B ring, significantly influence antioxidant activity [39]. In addition, it is known that the position and number of sugar units in flavonoids present an important role in antioxidant activity, and aglycones are more effective antioxidants than their corresponding glycosides [40]. The difference in ROS decrease between AA_770_ and isorhamnetin confirms this structure–antioxidant activity relationship.

### 3.6. AA_770_ Induces an Antiproliferative Effect on MDA-MB-231 but Is Ineffective on Cell Migration

As previously mentioned, ROS accumulation can affect cell migration and proliferation; thus, controlling their production may reduce cancer cells proliferation [7]. Furthermore, the antiproliferative action of isorhamnetin has already been described [41], as well as its ability to inhibit cancer cells migration [42]. Based on these findings and our previous results, we decided to investigate the effects of AA_770_ on the proliferation and migration of MDA-MB-231 cells. Cell proliferation was assessed over a 4-day period using the CCK-8 assay (Figure 6). A reduction in cell proliferation was observed 24 h after treatment with 10 µmol·L^−1^ AA_770_ and isorhamnetin, compared to the control condition (DMSO). This antiproliferative effect persisted over time, resulting in a 30% decrease in cell proliferation with AA_770_ and a 40% decrease with isorhamnetin after 4 days of treatment.

Cell migration was assessed in the presence of AA_770_ using the wound healing assay. The results indicate no significant difference in wound closure time compared to the control, suggesting that AA_770_ does not affect cell migration under the tested conditions (Appendix A Appendix A).

Even if it is well established that ROS levels influence many cellular processes including cell proliferation [43], it is difficult to conclude about ROS modulation by the tested compounds and their effect on cell proliferation. Indeed, flavonoids are known to act on cell proliferation by various mechanisms beyond their direct antioxidant activity. Notably, they can regulate crucial transcription factors such as activator protein 1 (AP-1) and nuclear factor κB (NF-κB) [44,45]. Additionally, they can function as epigenetic regulators, influencing gene transcription [46] but also demonstrate less specific activities, such as interacting with biological membranes.

Finally, the bioactivity of flavonoids is notably linked to their interaction with biological membranes. This interaction with membrane lipid bilayers influences their cell membrane permeability, a key pharmacological mechanism for antioxidant and antiproliferative phytochemicals [47], and the molecular weight, due to the presence or not of glycosyl groups, could reduce this cell permeability [48]. Also, previous studies reported that variations in the antiproliferative effect of flavonoids may be attributed to the presence of *O*-glycosylation, which usually reduces their effectiveness against cancer cells [39]. Our present findings support this structure–activity relationship, as the tested glycosylated compound AA_770_ presents a weaker bioactivity (antioxidant and antiproliferative activities) than the aglycon (isorhamnetin) form.

## 4. Conclusions

The present study is considered a first attempt to identify the compounds from *A. henoniana* stems methanol extract, as well as the determination of the biological potential of its purified compound (AA_770_). Using LC–MS/MS analysis, we identified a significant phytochemical, AA_770_, an isorhamnetin derivative with three glycosidic linkages. Our analytical approach, combining LC–MS/MS and NMR spectroscopy, allowed for a comprehensive structural and functional elucidation of AA_770_.

AA_770_ did not exhibit cytotoxicity against MDA-MB-231 breast cancer cells and displayed significant antioxidant activity by reducing ROS levels induced by TBHP, albeit to a lesser extent than the aglycone isorhamnetin. This diminished activity could be attributed to the presence of glycosidic moieties, which often reduce the antioxidant capacity of flavonoids compared to their aglycone forms. Finally, both AA_770_ and isorhamnetin demonstrated notable antiproliferative effects on MDA-MB-231 cells. This observation supports the existing literature, which suggests that flavonoid glycosylation generally diminishes antiproliferative activity. Over a 4-day treatment period, AA_770_ and isorhamnetin significantly inhibited cell proliferation, highlighting the potential of flavonoid derivatives in cancer therapy. Also, this study represents an initial attempt to investigate the AA_770_ natural compound, isolated and identified for the first time. In the future, bioactivities of AA_770_ should be assessed on additional breast cancer cell subtypes. To evaluate its broader efficacy, studies on animals are necessary to evaluate its effect in a more integrated physiologically relevant environment. Likewise, as for any potential cancer pharmaceutical agent, the effects of long-term exposure should also be assessed to evaluate putative cellular adaptative mechanisms.

The distinct biological activities of AA_770_ and isorhamnetin underscore the critical role of glycosylation in modulating flavonoid function. Glycosides tend to have higher molecular weights and different physicochemical properties, which can influence their interaction with cellular membranes and biological targets. The precise positioning of hydroxyl groups and sugar units significantly affects the antioxidant and antiproliferative effects, as demonstrated by the superior performance of the aglycone isorhamnetin in reducing ROS levels and inhibiting cell proliferation.

These findings highlight the nuanced role of glycosylation in flavonoid bioactivity and underscore the potential of such compounds in therapeutic applications, particularly in cancer treatment. Further research should focus on optimizing the glycosylation patterns to enhance the therapeutic efficacy of flavonoid derivatives.

## Figures and Tables

**Figure 1 antioxidants-13-00793-f001:**
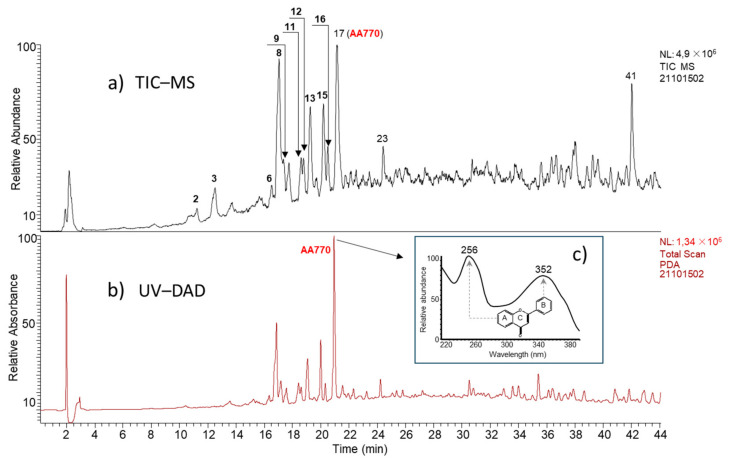
Chromatograms of *Anthyllis henoniana* methanol extract. (**a**): Mass chromatogram. (**b**): UV–DAD chromatogram. (**c**): Absorption of AA_770_ compound (retention time = 20.95 min).

**Figure 2 antioxidants-13-00793-f002:**
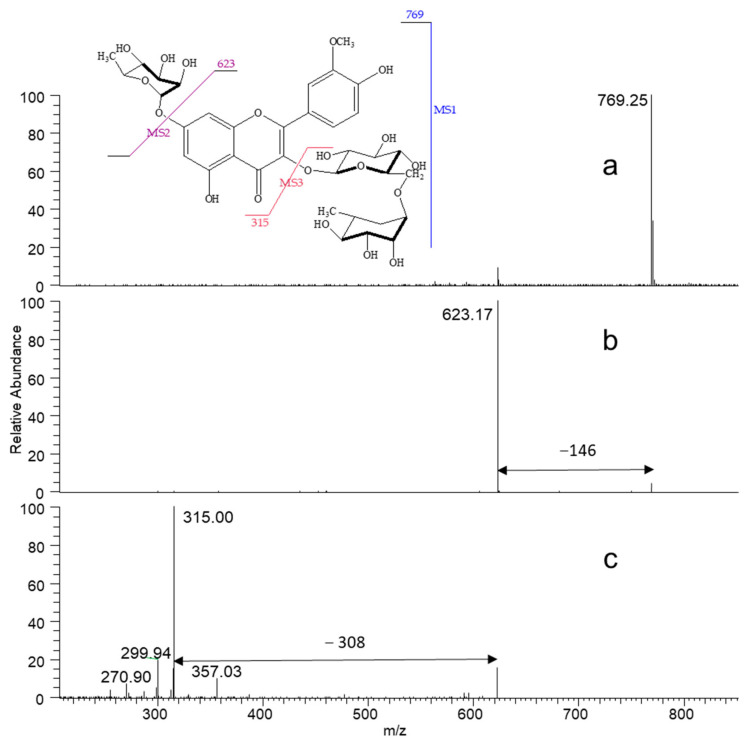
ESI–MS mass spectra of compound 17(AA_770_): Isorhamnetin-3-*O*-rutinoside-7-*O*-rhamnopyranoside: (**a**) full scan mass spectrum; (**b**): MS^2^ of 769, giving *m*/*z* = 623, corresponding to the loss of rhamnose; (**c**): MS^3^ of 769, fragmentation of 623.17 mainly generated the aglycone *m*/*z* = 315.

**Figure 3 antioxidants-13-00793-f003:**
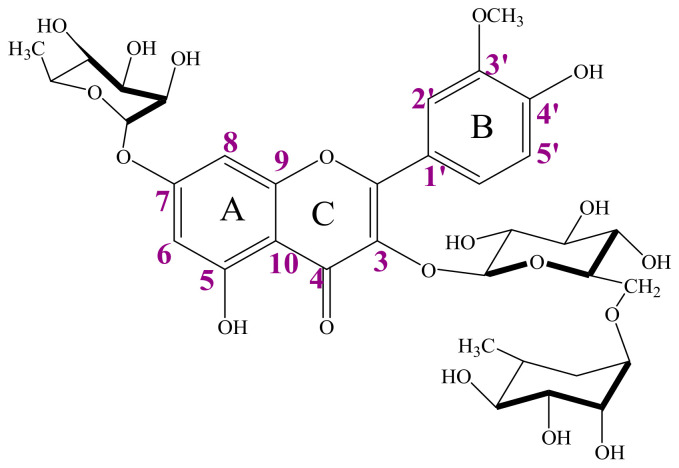
Structure of AA_770_: Isorhamnetin-3-*O*-rutinoside-7-*O*-rhamnopyranoside.

**Figure 4 antioxidants-13-00793-f004:**
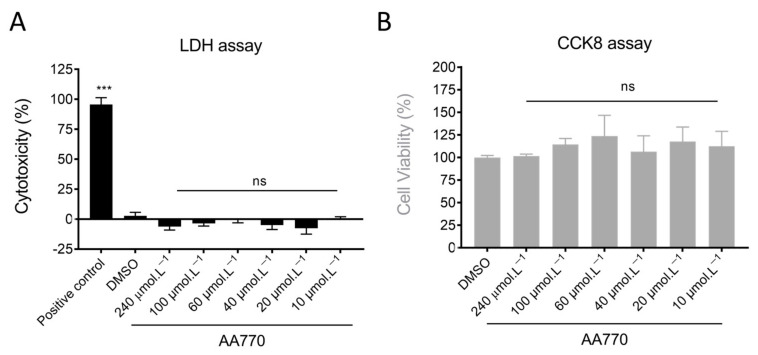
Effect of AA_770_ on MDA-MB-231 cytotoxicity and viability. Cytotoxicity of AA_770_ was tested by measuring cell mortality by LDH release (**A**) and viability by CCK-8 assay (**B**). Results are expressed as percentages with respect to the positive control for each assay: lysis buffer for LDH and DMSO for CCK8. The values are means ± SE of four independent experiments, each performed in triplicate. For the LDH assay, positive control is obtained with lysis buffer. In both assays, DMSO conditions were established by incubating the samples with 0.5% DMSO for 24 h. Means were statistically compared to DMSO (ns: not significant, *** *p* < 0.001 Tukey’one-way ANOVA test).

**Figure 5 antioxidants-13-00793-f005:**
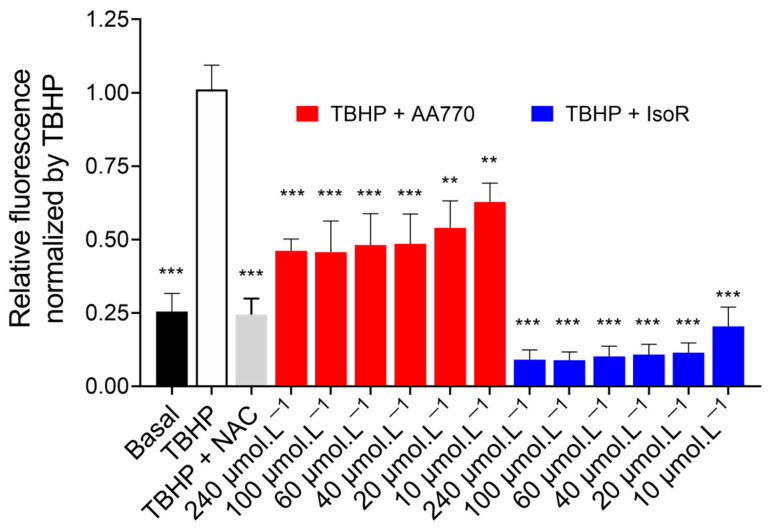
Evaluation of ROS production in non-treated MDA-MB-231 cells and cells treated with TBHP, AA_770_, and isorhamnetin (IsoR). Cells were incubated for 4 h with buffer (Basal, black bar) or buffer with TBHP at 30 µmol·L^−1^ plus 0.5% DMSO (white bar), 3 mmol·L^−1^ NAC (grey bar), or various concentrations of AA_770_ (red bar) or isorhamnetin (blue bar) as indicated. After incubation, ROS levels were evaluated using the 2′,7′-dichlorodihydrofluorescein diacetate assay. Histograms indicate the normalized fluorescence for each condition using TBHP as reference. The bars represent the means ± SE of three independent experiments, each performed in triplicate. Means were found statistically different compared to TBHP (** *p* < 0.01; *** *p* < 0.001 Tukey’s one-way ANOVA test).

**Figure 6 antioxidants-13-00793-f006:**
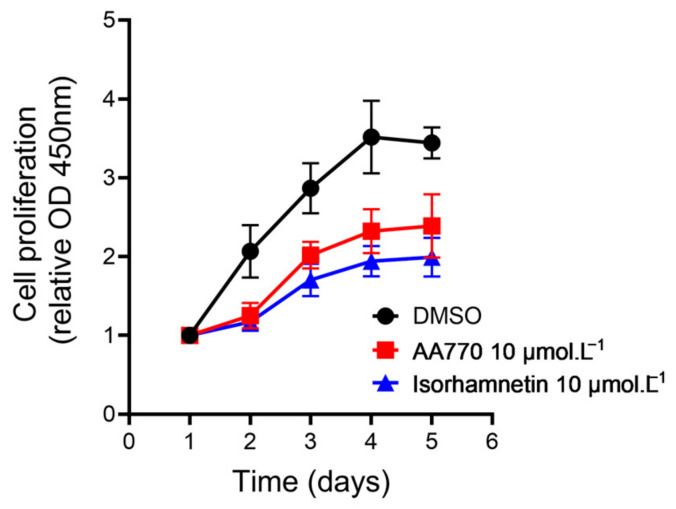
Effects of AA_770_ and isorhamnetin on MDA-MB-231 cells proliferation. MDA-MB-231 cells were treated daily with DMSO or 10 µmol·L^−1^ of isorhamnetin or AA_770_. Cell proliferation was measured using CCK-8 assay every day for 4 days after the beginning of the treatment. Results are presented as relative absorbance at 450 nM (mean ± SE, n = 3, each performed in triplicate).

**Table 1 antioxidants-13-00793-t001:** Extraction yields (%) of Anthyllis henoniana stems.

Solvent	Extraction Yield (%)
Hexane	1.1
EtOAc	3.6
MeOH	1.4

EtOAc: Ethyl acetate; MeOH: Methanol.

**Table 2 antioxidants-13-00793-t002:** Different phenolic compounds identified from *A. henoniana* methanol extract and their relative abundances.

*N*	Tr (min)	UV (nm)	RA (%)	*m*/*z*	LC/ESI–MS^2^ *m*/*z* (%)	MS^3^	Attribution	Ref
*1*	11.20	272	1.8	371	209/149		Caffeoylquinic acid derivative	[11]
*2*	12.04	260–376	0.2	611	449/431		Luteolin-*O*-diglucoside	[11,12]
*3*	12.59	272	3.0	293	179(100)/161/143/119/89		Unknown	-
*4*	15.45	280–380	0.4	771	625 (100)/447	301	Quercetin triglycoside (hexose-deoxyhexose-hexose)	[13]
*5*	15.82	272–380	0.7	755	609/446 (100)/301	300(100)/445/489/463/271	Quercetin rhamnosyl-rutinoside	[13]
*6*	16.50	272–380	1.2	755	609	429/285/257	Kaempferol rhamnosyl-diglucoside	[13]
*7*	16.70	272–332	0.5	885	739	593/575(100)/473/393/327/285/255	Kaempferol conjugate	[14]
*8*	17.05	272–328	13.9	593	575/503/473(100)/383/353	383/353(100)	Vicenin-2	[15]
*9*	17.36	250–340	1.8	915	769		Divarioside A	[16]
*10*	17.43	256–352	1.3	785	639		Isorhamnetin-*O*-rhamnoside-*O*-hexosyl-hexoside	[17]
*11*	18.58	272–332	3.7	563	545/503/473(100)/443/383/353	353/383	Isoschaftoside	[18]
*12*	18.80	272–380	3.2	755	609	429/285/257	Kaempferol rhamnosyl-diglucoside	[19]
*13*	19.25	256–348	8.0	563	545/503/473/443(100)/383/353	353/383	Schaftoside	[18]
*14*	19.65	276–380	1.6	415	149/233/293/191		Alpinoside	[20]
*15*	20.19	264–344	5.9	739	593	285(100)/327/255	7 Kaempferol-3-*O*-(Hexose-Rhamnoside)-7-*O*-deoxyhexose	[21]
*16*	20.52	272–380	3.2	563	545/503/473/443(100)/383/353	353/383	Schaftoside	[18]
*17* ^π^	21.17	256–352	18.5	769	623	357/315(100)/300/271	Isorhamnetin glucosyl-di-rhamnoside	[13]
*18*	21.78	268–324	1.7	609	343/301		7-*O*-rhamnogalactoside quercetin	[22]
*19*	22,52	224–324	1.6	463	301		Isoquercetin	[15]
*20*	22.81	284–380	0.4	597	579/477/417/387/357	315(100)/239	Phloretin-3′,5′-di-*C*-glucoside	[23]
*21*	23.44	280–380	0.7	533	515/473/443(100)/383/353		Apigenin-*C*-pentoside-*C*-pentoside	[17]
*22*	23.86	280–380	0,3	523	361(100)/347		Secoisolariciresinol hexoside	[24]
*23*	24.42	252–340	3.0	623	315/300/271/255		Isorhamnetin-3-*O*-rutinoside	[25]
*24*	25.24	224–328	0.4	477	315(100)/357/449/300/151		Isorhamnetin 3-*O*-galactoside	[26]
*25*	25.54	224–324	1.5	463	301		Isoquercetin	[15]
*26*	25.76	224–324	0.3	595	551/475/445/343/301/257/191		Quercetin-3-*O*-apiosyl-(1→2)-galactoside	[27]
*27*	26.00	280–380	0.7	577	283/268		Lanceolarin	[28]
*28*	28,2	228–280	0.3	581	287/449/471		Cyanidin-3-*O*-sambubioside	[12]
*29*	30.94	228–340	0.8	373	355 (100)/337/301/263/151		7-Hydroxymatairesinol	[29]
*30*	31.76	228–284	1.2	299	271/284/255/240		Rhamnocitrin	[30]
*31*	32.79	228–316	0.6	593	447/285		Kaempferol-3-*O*-rutinoside	[15]
*32*	36.29	228–288	2.7	271	177/151(100)		Naringenin	[31]
*33*	37.04	232–284	0.9	327	309/291/229(100)/211/209/171		*Oxo*-dihydroxy-octadecenoic acid	[13]
*34*	37.37	284	1.1	285	285/257/241/217/199/189/175/151/1252/107		Luteolin	[32]
*35*	37.46	280	0.7	299	284		Kaempferide	[33]
*36*	37,75	280	2.0	283	268		Acacetin	[34]
*37*	38.87	228–288	1.4	351	336 (100)/323/308		Hedysarimpterocarpene B	[35]
*38*	39.62	232–304	1.6	329	311/293/229(100)/211/171		Tricin	[36]
*39*	40.55	236–280	0.7	357	339/285(100)/151/109		Unknown	-
*40*	41.02	236–360	2.0	255	135(100)/119/91		Liquiritigenin	[37]
*41*	42.04	228–344	4.0	313	298	283(100)/270/255	Chrysoeriol-methylether	[32]

RA (%): Relative abundance. ^π^: Main compound to be purified for biological activities.

## Data Availability

Data is contained within the article.

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
