# Peer review of "Isolation, NMR Characterization, and Bioactivity of a Flavonoid Triglycoside from Anthyllis henoniana Stems: Antioxidant and Antiproliferative Effects on MDA-MB-231 Breast Cancer Cells"

_antioxidants, 2024, doi:10.3390/antiox13070793_

Round 1

Reviewer 1 Report

This manuscript describes the characterization of a methanolic extract of Anthyllis henoniana, the elucidation of the structure of a novel compound and the evaluation of its cytotoxicity, and of its modest antixidant activity in living cells. 

The contents are of limited interest, however the paper could be published after major revisions. I report here the point already raised in the recommendations, with further comments:

1. The protocol for the extraction is poorly described, the volumes of the solvents are not reported, the times of soxhlet extractions are not reported, the amount and yields of the extracts are not reported. It is very importat to report such details in order to allow to reproduce the results and to evaluate the yield of the potential bioactive compound also from the economic point of view.

2. In cytotoxiciy assay and other cellular tests, the amount of dmso in the final solutions is not reported.

and:

2'. On discussing the cytotoxicity, the authors claim that it is not statistically different from that of dmso, however no statistic data are given for dmso at higher concentration (error bar is not present in figure 4). Moreover, no ANOVA test has been perfomed in order to give evidence of the claim.

It is really unclear to me if dmso has been used as a co-solvent (in large molar excess) or at the concentrations reported in the discussion on cytotoxicity. Anyway, the main point is that the test with dmso seems not replicated.

3. in section 2.10, statistics, the first sentence is meaningless: Results are the means SE of the number (n) of observations. Maybe you mean: are the means +/- the SE of the number of observations?

4. All the NMR spectra described in the results MUST be reported, at least as supplementary materials in order to allow the evaluation of the spectra quality and to give a reference to the reader. 

The discussion on NMR is very difficult to read in the absence of the spectra, and the purity of the compound can not be assessed.

Finally, I believe also that in case of publication, the title of the paper should be mtigated in some way, as it generates an exaggerated expectation in the reader.

key reference 3 is reported in an incomplete way

Reviewer 2 Report

The study explores the biological activity of AA770, a triglycosylated derivative of isorhamnetin, focusing particularly on its effects on the MDA-MB-231 triple-negative breast cancer cell line. The research highlights AA770's lack of cytotoxicity, moderate antioxidant activity, and significant antiproliferative effects compared to isorhamnetin's stronger activity profile. These findings contribute valuable knowledge to the ongoing investigation of natural compounds with potential therapeutic benefits for breast cancer.

However, there are several areas where the study could be improved:

-The rationale for selecting breast cancer cells, specifically a TNBC cell line, should be clearly justified. TNBC is characterized by aggressive behavior and limited treatment options compared to other breast cancer subtypes. Please include in the text why this specific subtype was chosen.

-Furthermore, focusing on a single breast cancer cell line limits the generalizability of the results. Including multiple cell lines, particularly representing other subtypes of breast cancer, could help assess the broader efficacy of AA770. Have preliminary data from other cell lines been considered?

-Moreover, while the exclusive use of in vitro methods provides initial insights, it does not fully capture the complexities of in vivo systems. Future studies should incorporate in vivo animal models to validate the observed effects of AA770 and explore its potential translational implications. This aspect should be addressed in the discussion to contextualize the findings within the broader scope of cancer research.

-While the study provides evidence of antioxidant and antiproliferative activities, it does not deeply investigate the underlying molecular mechanisms. Understanding the specific pathways and molecular targets involved could provide more comprehensive insights into how AA770 exerts its effects.

-While the study uses two concentrations of AA770, a more extensive dose-response analysis could identify the optimal therapeutic window and better characterize the compound's efficacy and safety profile.

-The exposure period for AA770 and isorhamnetin was relatively short (24h for cytotoxicity and up to 4 days for proliferation). Long-term effects, including potential chronic toxicity or adaptive cellular responses, were not assessed. The authors should discuss that future research should consider chronic exposure.

-The study utilized the DCFDA assay to measure ROS levels. While this method is useful, it has limitations and can sometimes produce artifacts. Employing additional methods to measure ROS would strengthen the findings.

-While the study notes that glycosylation can influence antioxidant activity, it does not thoroughly explore how different glycosylation patterns affect the bioactivity of AA770. A more detailed analysis of different glycosylated derivatives could provide a clearer understanding of structure-activity relationships.

Round 2

Reviewer 1 Report

all my comments have been addressed, in my opinion the paper can be published

all my comments have been addressed, in my opinion the paper can be published

Author Response

Thnak you for your comment and your review

Reviewer 2 Report

The authors have addressed the concerns I raised.

However, I believe it is important for them to state that this study represents an initial attempt to investigate this natural compound. This clarification, in line with their responses to the comments, would provide a justification for why the research does not extend further at this stage.

-

Author Response

Thank you for your comments.

We add two sentences to point out that this is the start of the AA770 compound study.

One is at the end of the abstract: This study provides a preliminary investigation into the pharmacological potential of the natural compound AA770 isolated and identified from Anthyllis henoniana for the first time.

The other in the conclusion (line 378-379): Also, this study represents an initial attempt to investigate the AA770 natural compound, isolated and identified for the first time

With the other sentences about the remaining work on the compound mecanistic, we think the message is clear.